# Intelligent Control of Mushroom Growing Conditions Using an Electronic System for Monitoring and Maintaining Environmental Parameters

Žydrūnas Kavaliauskas *, Igor Šajev, Giedrius Gecevičius and Vytautas Čapas

Department of Industrial Engineering and Robotics, Kaunas University of Applied Sciences, Pramones Ave. 20, LT-50468 Kaunas, Lithuania
* Correspondence: zydrunas.kavaliauskaslei@gmail.com

**Abstract:** In order to make plant cultivation as economical and efficient as possible, and to reduce the amount of human errors and labour costs, autonomous systems for growing plants and, in this case, also mushrooms, are increasingly being implemented. In this work, a prototype of an electronic/automatic system was designed and manufactured for the monitoring and control of the parameters of mushroom growing conditions. Appropriate sensors were used to monitor the application parameters such as $CO_2$ level, temperature and humidity, and the data were sent to the main logic controller using an RS 485 interface-based local data transmission network. Both the main controller and the individual parts of the system were made on the basis of PIC18F25K83 microcontrollers, using the C programming language to create the program code. In order to achieve optimal system operation, a software algorithm was created to ensure fast system operation. During the production, the PCBs of the system were optimized to achieve the smallest possible dimensions and optimal construction and the arrangement of the active electronic elements. For the convenience of the user, a system application was created so that it is possible to monitor information from the environmental sensors and the process of parameter control. This work aimed to show that such autonomous systems based on environmental sensor monitoring are universal and can be applied to a large number of plant species. In addition, the capabilities of the plant cultivation system can be expanded if needed by additionally connecting relevant environmental sensors and environmental parameter regulation units.

**Keywords:** smart cultivation control; electronic cultivation control; environmental parameters control; plant cultivation control; precision agriculture; environmental sensor monitoring; mushroom cultivation control

## 1. Introduction

Recently, more and more attention has been paid to increasing work efficiency and to reducing costs. In addition, great attention is paid to reducing service staff. These aspects of increasing the efficiency of the production process are becoming more and more important in agriculture, in the cultivation of various food crops, etc. In order to cope with today's challenges, various technological solutions based on innovation are needed [1–4]. One of such areas, that requires innovative solutions for growing and monitoring envi-ronmental parameters, is the cultivation of various edible mushrooms. Mushrooms are grown industrially in various types of premises (greenhouses, hangars, etc.), where it is technically and technologically possible to ensure the necessary conditions for the growth process. In order to optimize the growth process of the mushrooms, it is necessary to en-sure both observation and monitoring of indoor environmental parameters, as well as control of environmental parameters. In mushroom cultivation, it is important to monitor such environmental parameters as $CO_2$ concentration level, humidity, temperature, etc. [4–8]. In order to monitor various environmental parameters, various types of sensors

are needed, which collect information and send it to a central computer or microcontroller. Sensors can be of various types: 1. Optical sensors are often used in drone or other systems to determine soil moisture; 2. Electromagnetic sensors that can be inserted into the soil are used to measure soil pH, salinity level, etc.; 3. Electrochemical sensors that are used to determine various ions such as $K^+$, $H^+$, etc.; 4. Location sensors such as GPS are needed to ensure efficient fertilization of fields; 5. Air flow sensors are used to measure the amount of air in the soil; 6. Acoustic sensors are used, e.g., to detect pests in the mushroom growing premises [9–11].

Both the microcontroller and the computer analyse the information received from various sensors and make appropriate decisions on how one or another environmental parameter can be adjusted (e.g., temperature, humidity, etc.). Environmental parameter correction decisions are made either by a dedicated program on a computer or microcontroller, or by the operator of this system. Decision-making related to the management and correction of environmental parameters depends on the architecture of the electronic sys-tem. Artificial intelligence is increasingly used for parameter control and decision-making. Such a system makes it possible to significantly reduce the probability of errors in the control of growing conditions and environmental parameters. Such systems can be completely autonomous and operate without human intervention for a long time. At the same time, in order to control the parameters and make corrections, not only the sensor system is needed, but also various devices (water pumps to maintain humidity, heaters and recuperators to maintain temperature, etc.) [12–15]. The control of devices that ensure the control of environmental parameters can be carried out by the same system, which simultaneously processes sensor information and makes appropriate decisions related to changing environmental parameters. The control of various devices can be carried out using relay circuits or with the help of powerful transistors (e.g., MOSFETs) by cutting off or turning on the power supply at certain moments of time. Industrial controllers, such as Siemens, can be used to control devices and analyze information from sensors, and also electronic circuit-based programmable microcontrollers such as PIC or Arduino [16–18]. In order to simplify the electric circuits and make their work as simple as possible, optimization tasks are often solved, allowing to reduce the production costs of control systems and helping to ensure their stable operation. At the same time, it is important not only to monitor the environmental parameters, but to make their corrections at the right moment and ensure uninterrupted power supply. In order to avoid power fluctuations in power circuits or microbreaks, backup power systems such as UPS or high-capacity (1 F and more) capacitor batteries are often used [19–22].

Information related to environmental parameters, recorded by sensors (humidity, temperature, etc.) and converted into a digital or analog signal, respectively, can be sent to the central system in various ways: IoT technology, RF local radio receiver–transmitter network, RS 485, RS 232, Ethernet network, etc. Programmable microcontrollers are widely used to ensure the process of growing mushrooms, because they are characterized by high reliability, low cost, low risk of failure, and, as stated in the literature, they are characterized by good resistance to environmental effects [23–25]. Control systems based on microcontrollers are characterized by small geometric dimensions, which is important for saving space, and microcontroller systems can be powered by relatively low-power (10 W and more) power supplies. The microcontrollers of the PIC series are also convenient that they have a sufficient number of logical inputs and outputs, allowing both to collect information sent by various sensors and to control various devices (for example, turning water pumps ON/OFF at the right time, etc.). While using the microcontrollers, it is possible to reliably ensure the limits of fluctuations in environmental parameters and track their deviations from the norm (e.g., ambient temperature, etc.). Cloud technology or a database such as MySQL can be used to store the operating modes of the devices and the information received from the sensors. The control system can be connected to the data cloud using IoT technology, while the database can be connected to the control system using a local area network such as Ethernet, etc. The main problem of the sensor systems

discussed in the literature is that most of the attention is paid to the sensors themselves, but little attention is paid to the universality of systems and stability of data transmission of individual parts [1,26–28].

The automatic environmental parameter control systems presented in most literature sources are not universal and are only suitable for certain types of mushrooms. The aim of this work is to create a universal automatic system for maintaining environmental parameters (using temperature, humidity and $CO_2$ level sensors and a hybrid data transmission network) with an application that ensures the cultivation process of various types mushrooms (as well as other plants).

## 2. The Experimental Setup and Methodology

An electronic system based on PIC microcontrollers was developed to monitor and control the environmental parameters and growth conditions of the mushroom growing room. The monitored and controlled growth environment parameters are humidity, $CO_2$ level and ambient temperature. The mushroom growing premises consisted of 2 rooms in which 4 humidity and temperature measurement modules (sensors) and one $CO_2$ level measurement module (sensor) are arranged.

Air conditioner, recuperator and humidifier units are used to maintain the climate. The control of these units for maintaining environmental parameters is carried out with the help of a microcontroller, taking into account the information sent by the sensors in the room and the mushroom cultivation algorithm. The general concept of the system for maintaining the environmental conditions for mushroom cultivation is presented in Figure 1.

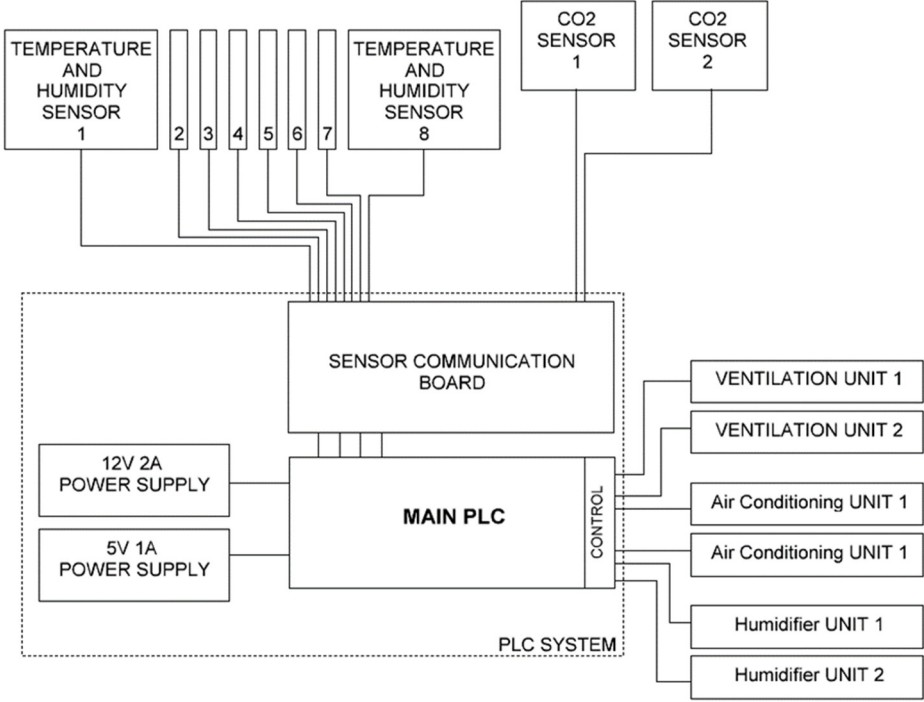

**Figure 1.** General conceptual diagram of the automatic control system of environmental parameters.

The PLC controller system consists of two PCB boards: (1) MCU, control relays and control MOSFETs, input communication module board for signal transmission using optocouplers; (2) central communication board for remote humidity, $CO_2$ and ambient temperature modules. The SHT41 module was used to record temperature and humidity. The SHT4x builds on a completely new and optimized CMOSens® chip that offers reduced power consumption and improved accuracy specifications. With the extended supply voltage range from 1.08 V to 3.6 V, it is the perfect fit for mobile and battery-driven applications.

The high-accuracy versions SHT41 and SHT45 feature typical and maximal accuracies that have been honed even further, down to $\Delta RH = \pm 1.5\%$ RH and $\Delta T = \pm 0.1$ °C. The electrical power of the temperature and humidity module is 0.05 W, the voltage is 5 V. The temperature and humidity measurement system consists of two PCB boards: (1) communication board with the main PLC controller; (2) mini board of temperature and humidity sensor SHT41. An AR257 meter was used for $CO_2$ measurement, the measurement limits of which are 0–10,000 ppm, the temperature measuring range 0–50 °C, power supply 2.5 W, voltage 24 V. The $CO_2$ measurement system consists of: (1) communication board with the main PLC controller; (2) industrial $CO_2$ meter (sensor) AR257. $CO_2$ levels during system testing both before and during mushroom cultivation were kept lower (about 270–275 ppm depending on test conditions) than fresh outside air ($CO_2$ levels in fresh air vary between 300–450 ppm depending on location). This process was ensured by the room's two-way ventilation system, which consisted of two ventilators with a diameter of 50 cm. One ventilator removed the room air with excess $CO_2$ gas, while the other ventilator blew in fresh air from the environment. Air blown in from the environment first entered the ZPure $CO_2$ filter, thus producing a lower $CO_2$ level indoors than in fresh air. The ZPure $CO_2$ filter removes carbon dioxide from inert gases, He, Ar, $N_2$, $H_2$, and clean dry air (CDA) to low ppb levels. The filter functions by consuming carbon dioxide in a reaction with highly dispersed NaOH on a silicate support.

The environmental parameter control system for mushroom growing premises is developed using a PIC18F25K83 microcontroller. The PIC18(L)F25/26K83 microcontroller family with CANTM technology in 28-pins features a 12-bit ADC with computation (ADC2) automating capacitive voltage divider (CVD) techniques for advanced touch sensing, averaging, filtering, oversampling and threshold comparison. The C programming language was used for programming the microcontroller, while the control application was developed using the Delphi programming language.

### 3. The Results and Discussion

A prototype electronic system was developed for monitoring and controlling the environmental conditions of mushroom growing premises. The main PLC consisted of two stages: the main electrical circuit for control and data analysis and the communication circuit.

A PIC18F25K83 microcontroller was used to monitor and analyse the environmental parameters and to control the work of the units for maintaining the environmental parameters. The main electrical circuit of the PLC based on the aforementioned microcontroller is presented in Figure 2. The mushroom growing premises consisted of two rooms in which it was necessary to ensure the control of environmental conditions. In order to control the environmental conditions in the two rooms, the PLC electrical circuit included two inputs each for the control of air conditioners and fans. At the same time, information was received about the humidity in the premises. To control this process, the PLC electrical circuit had two inputs for the control of humidifiers. Since the control of environmental parameters devices is performed based on the information received from the environmental sensors, an RS 485 interface module is used to receive information from these sensors. The information packet travels digitally from the PLC communication PCB board to the RS 485 module.

A separate electrical circuit for communication based on the PIC18F25K83 microcontroller was developed and used to collect and transmit data from humidity, temperature and $CO_2$ sensors to the main microcontroller (which performs control and analysis of environmental conditions based on the information received from the sensors). The communication electrical circuit diagram is presented in Figure 3.

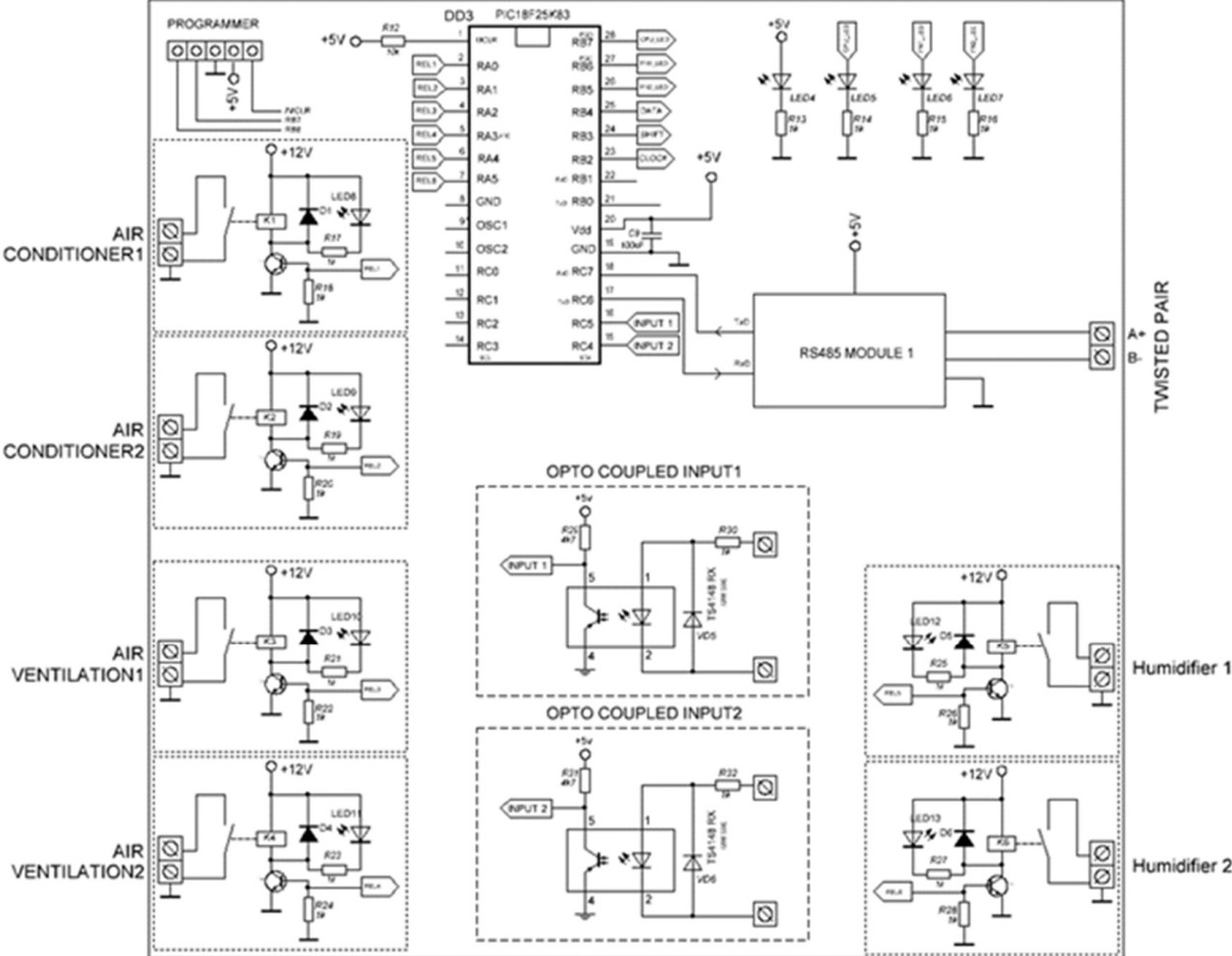

**Figure 2.** Main PLC electronic circuit.

The communication chain technologically has two inputs for collecting information from industrial $CO_2$ sensors (two $CO_2$ sensors since there are two mushroom growing rooms) [1]. There were four temperature and humidity measuring sensors in each room (SHT41 sensors are used to measure temperature and humidity), so the communication circuit has eight inputs for receiving information in digital form. An RS 485 interface is used to transfer information to the main microcontroller.

Separate electronic modules were made using PIC18F25K83 microcontrollers to convert the information of each humidity and temperature sensor into digital form and ensure the transmission process to the communication circuit of the main controller. The electrical schematic diagram of this module is presented in Figure 4. As we can see in the electrical circuit there is a humidity and temperature sensor input and a $CO_2$ sensor input. A total of eight units (to collect and process the data of eight humidity and temperature modules and two $CO_2$ modules) of such electronic circuits/PCBs were manufactured and four installed in each of two mushroom growing rooms. Using the RS 485 interface, the information from these sensors is transmitted to the communication board of the main controller.

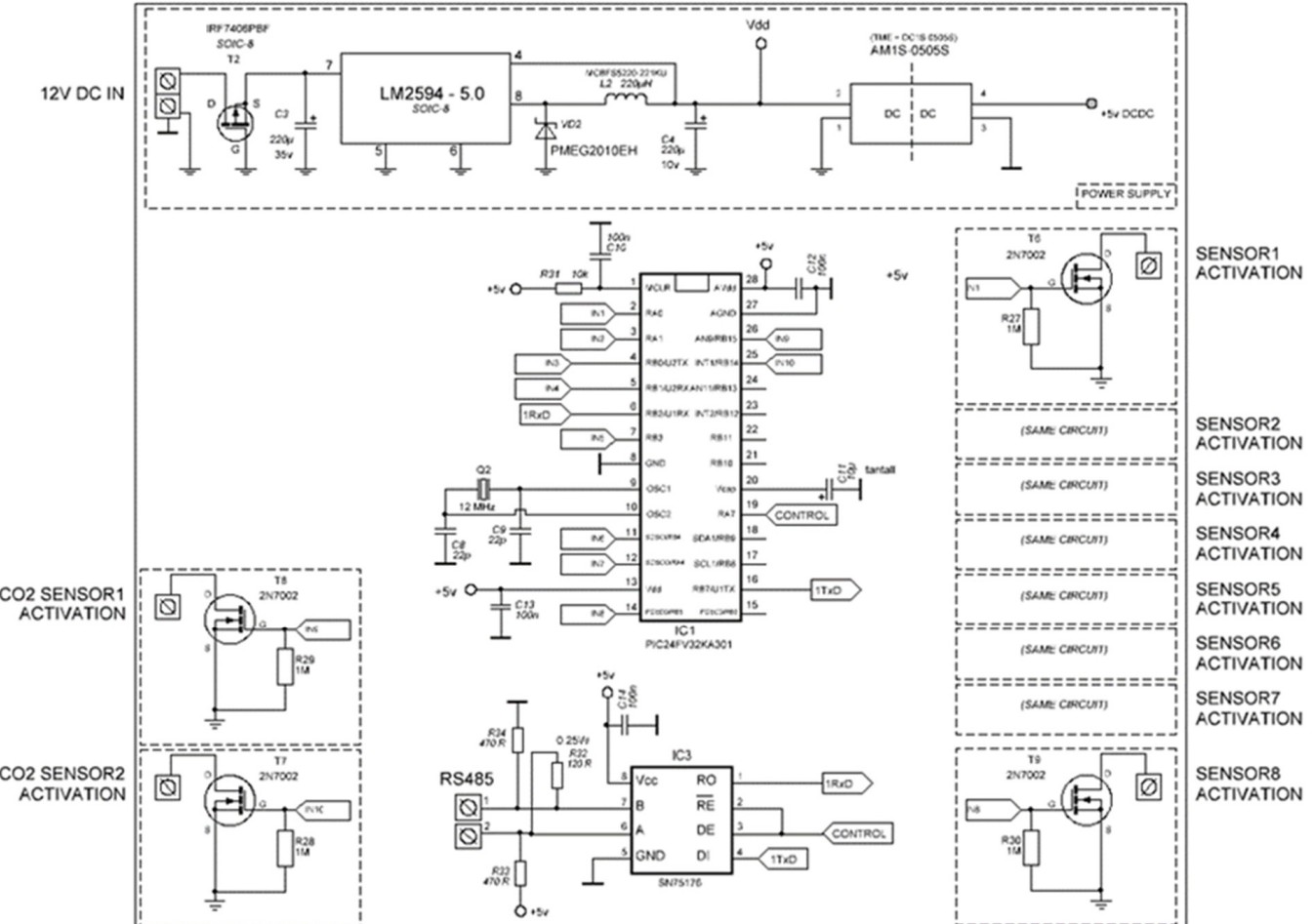

**Figure 3.** Main PLC communication power plant circuit.

The software for the microcontroller of main controller and communication microcontroller was written using C programming language. Using this language, a program was also written for individual electronic modules (Figure 4 electrical diagram) to collect information from humidity and temperature and $CO_2$ sensors [29,30].

The software algorithm of the entire mushroom cultivation conditions parameter measurement and control system is presented in Figure 5.

The program starts with the command "Start program". Later, the status of the environmental sensors is checked, whether the sensors are active, and the values of these sensors are recorded. Requests are sent to the electrical circuits of the sensors and checked for a response. After receiving information from humidity, temperature and $CO_2$ sensors, data analysis is performed and comparison is made to see if the environmental parameters are appropriate and correspond to the mushroom growth conditions [3]. After performing the analysis and receiving information from the sensors about deviations of the environmental parameters (temperature, humidity and $CO_2$), the regulators of these parameters are turned on or off (using MOSFETs and relay circuits), respectively, air conditioner, fan or humidifier. The status information of environmental sensors and environmental parameter maintaining devices is transmitted to the central computer, which also houses the system management application. Later, it is returned to the initial state of the algorithm "Start program" and the process is repeated.

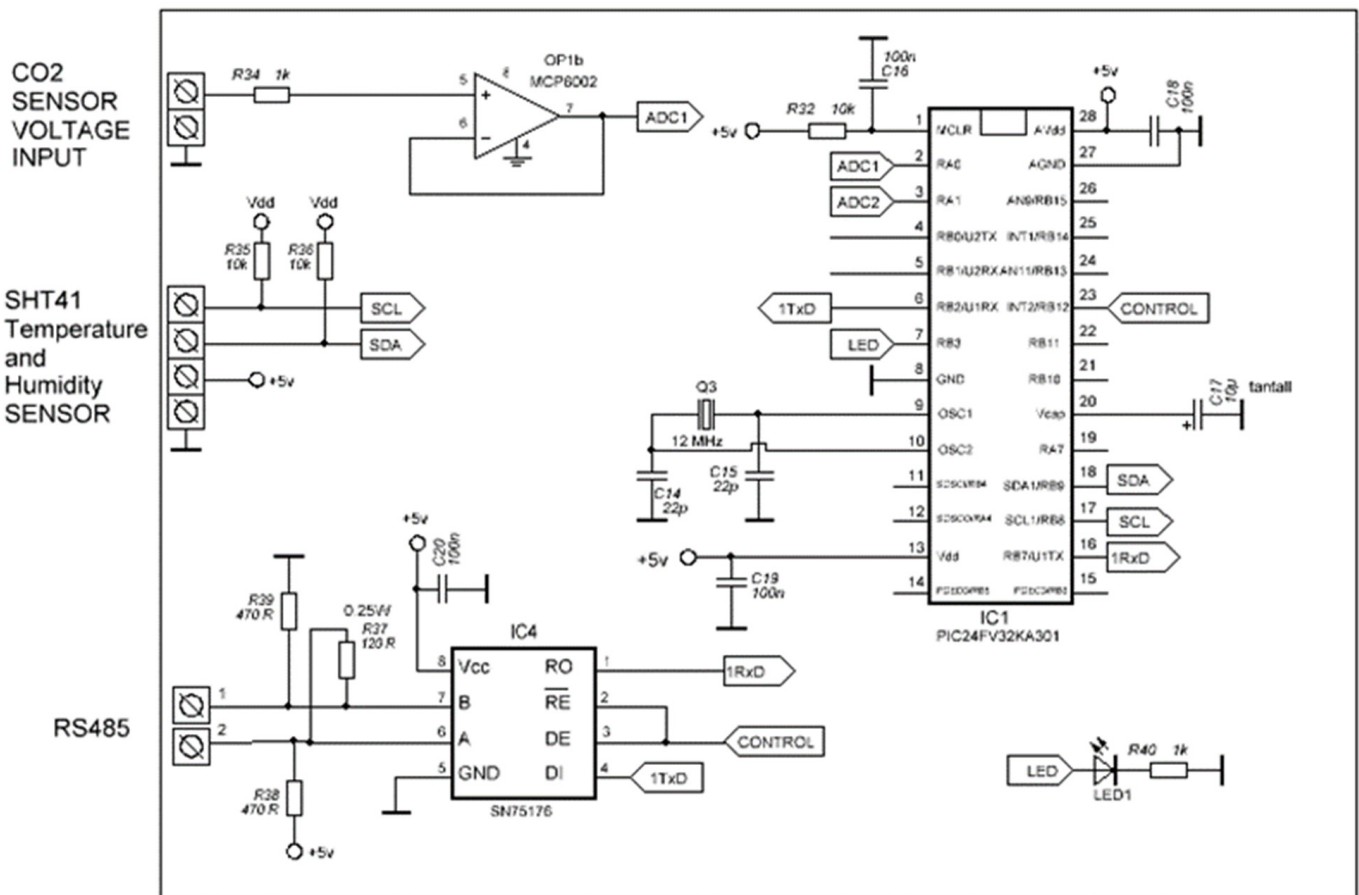

**Figure 4.** Sensor communication circuit.

Since both the sensors and the environmental parameters maintenance devices are located in two rooms, a data transmission network is required. The conceptual diagram of the data transmission network between the sensors and the PLC communication board is presented in Figure 6. This diagram shows how digital information is transferred between the sensor control modules and the central controller communication board. 5 V level digital signals are used to transmit information packets, while 12 V level digital signals are used for request-excitation [6]. A twisted pair of wires is used to send these packets. RS 485 interface modules are used to receive and send information packets between individual modules.

The general scheme of the data network is presented in Figure 7. The complete set of the data network consists of a PLC communication board, control modules for $CO_2$ sensors (two units), temperature and humidity sensors (eight units). RS 485 interface modules installed on all system PCBs are used to ensure the entire operation of the data transmission network.

A software application was developed using the programming language Delphi to ensure the monitoring and control of the environmental parameters of the mushroom growing premises. The main graphic window of system application management is presented in Figure 8. In the main window of the application, you can see the values of the environmental parameters of the two mushroom growing rooms, which are recorded by the environmental sensors. In the main window of the management application, the ON/FF statuses of environmental parameter control devices (e.g., air conditioners, etc.) can be monitored. In order for the entire mushroom growing conditions control system to work in autonomous mode, the "auto" mode must be selected in the control application.

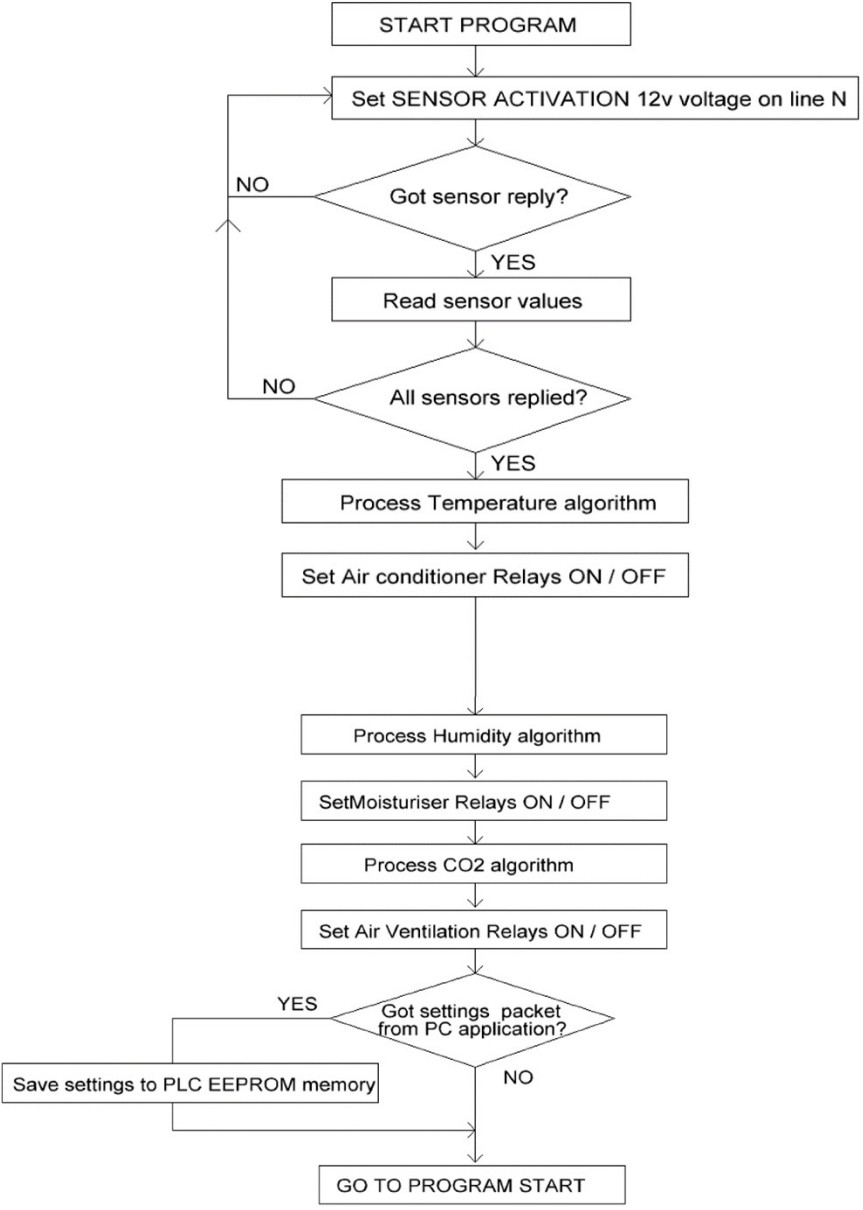

**Figure 5.** The algorithm of the program of the automatic system of maintaining conditions for growing mushrooms.

The conditions for growing the mushrooms depend on the type of mushrooms and the age of the mushroom plants. It is important to determine and maintain the appropriate environmental conditions when growing the different species of mushrooms. The control application has a program window for setting the operating values of devices that support environmental parameters (such as air conditioners, etc.) (Figure 9). In the window of the program, the boxes to change the parameters of the devices for maintaining the environmental conditions in the rooms to grow the mushrooms can be seen. After setting the required parameter values, in the main window of the application the autonomous mode can be selected, then the system can operate for the desired period of time without additional intervention of an operator. There are also possibilities to save the values of the set parameters so that they can be known in the future. In the control application, it is possible to record the average values and perform a reboot of the entire system.

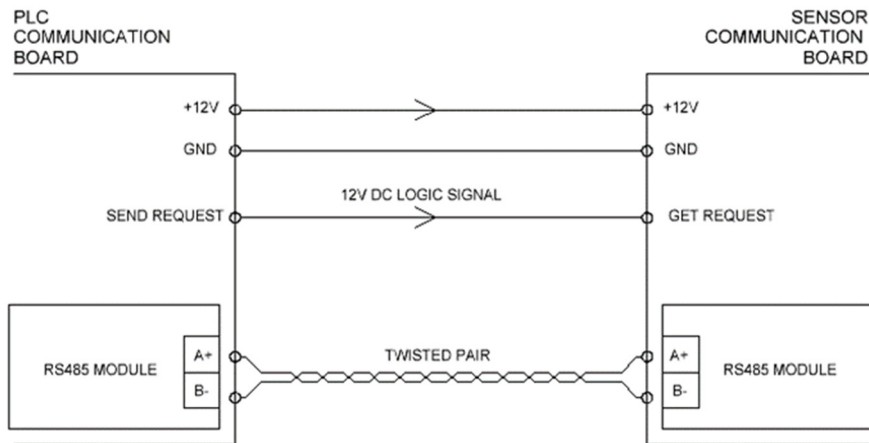

**Figure 6.** The diagram of data network communication between the sensor modules and the communication board of the main controller.

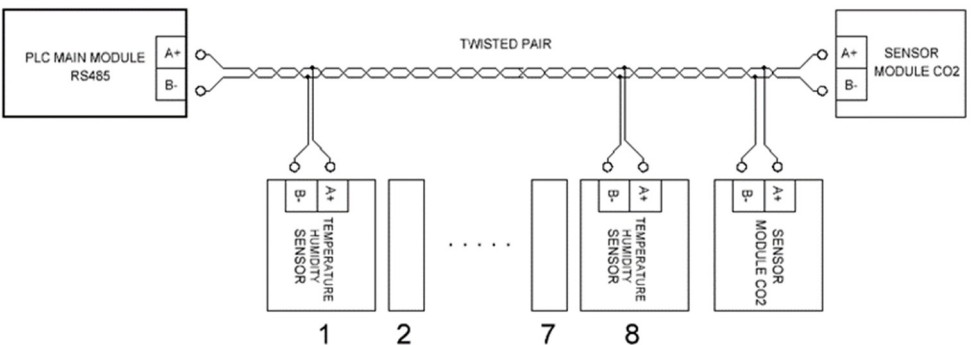

**Figure 7.** Network configuration diagram.

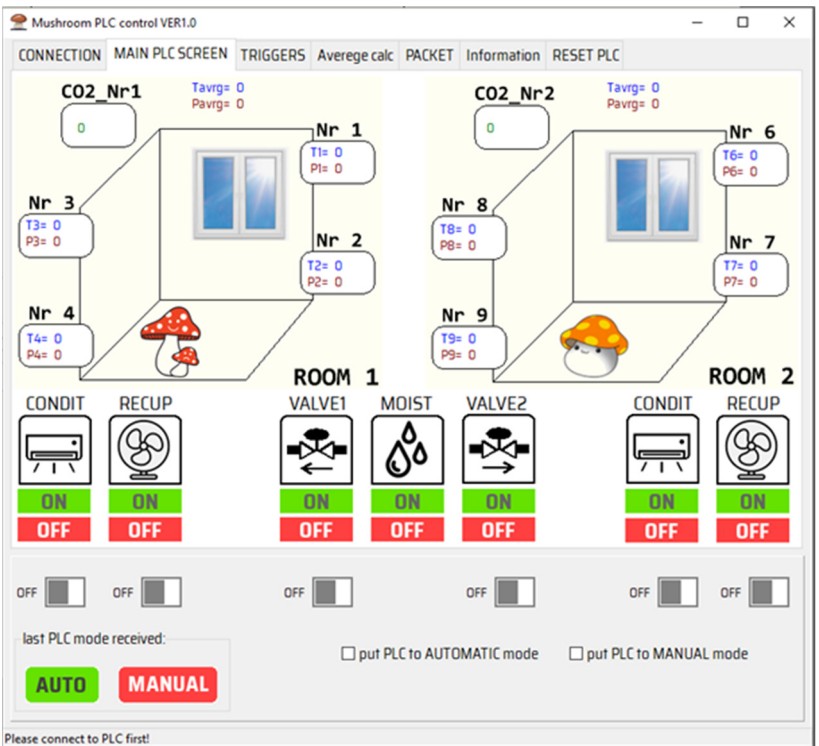

**Figure 8.** Management application to control the mushroom growing conditions.

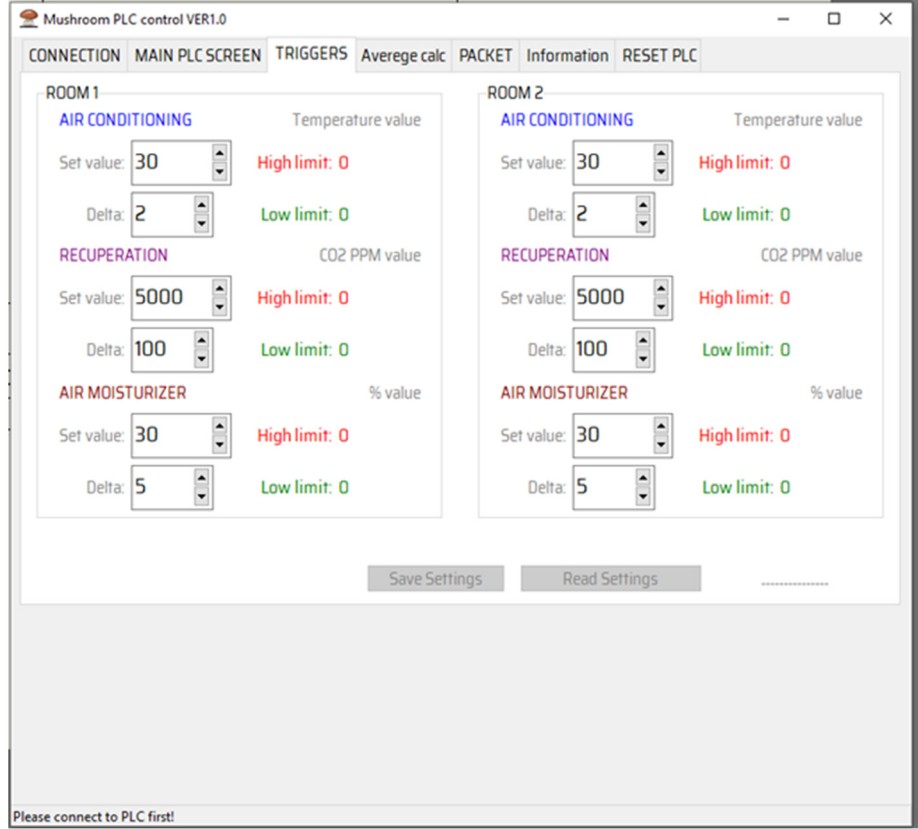

**Figure 9.** Values of parameters in the control application of mushroom cultivation conditions.

The printed boards were created for the realization of the prototype of the system of environmental parameters for mushroom cultivation. Figure 10 shows the main controller, which consists of the main microcontroller board and communication boards. Both of these PCBs are realized using double-sided printed circuit board manufacturing technology [1].

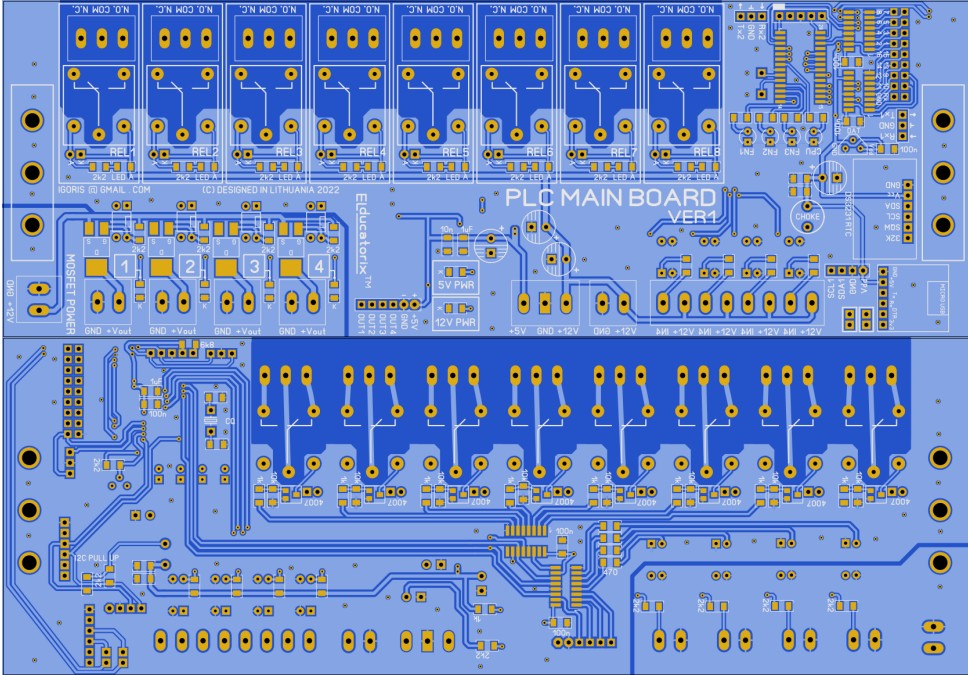

**Figure 10.** PCB view of the main electronic circuit of the controller.

Printed circuit board tracks are made of copper layer. In addition to the microcontroller, the printed circuit board was designed with spaces for active SMD elements such as capacitors, diodes, etc. In order to obtain a PCB with the smallest possible dimensions, PCB design optimization was carried out, allowing to reduce the number of tracks and make them as short as possible. A special dialectical layer is used to protect PCB active tracks from environmental effects, which is characterized by protection from both environmental factors and mechanical damage to the tracks.

The general view of the main controller and the communication board can be seen in Figure 11. The standard DIN rails were used for PLC installation, which are often used in electrical module installation processes. As it can be seen, press-on terminals are used for communication cable contacts.

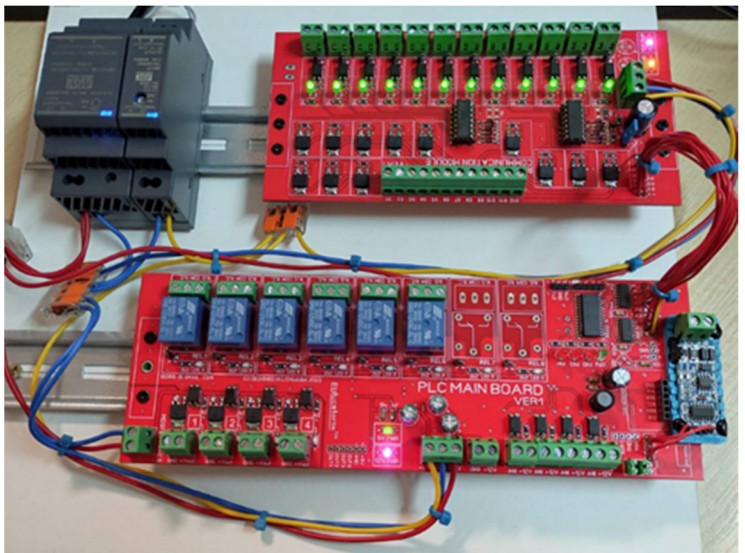

**Figure 11.** The general view of the main controller of the automatic control system.

A standard 100 W power supply (seen in the PLC photo) was used to power the PLC, which also has special brackets for mounting on the DIN rails. LEDs were used to display the information accordingly. Since the main PLC performs the function of powering, switching on/off circuits for environmental devices, galvanic decoupling using optocouplers has been used to protect certain parts of the PLC circuit from voltage surges (such as the microcontroller, etc.).

After the installation of the system for maintaining the parameters of the mushroom growing rooms, the work of this system was monitored in a trial mode. Test results were monitored over a 24 h period (Figure 12). As can be seen, the room temperature changes over time within the limits of about 0.2 °C, while the humidity level managed to be controlled within the limits of 0.2%. After analyzing the readings of the $CO_2$ sensor, it became clear that the level of this gas in the room changes within the limits of about 2 ppm. The conditions of the environmental parameters of the mushroom growing room can be changed accordingly using the developed control application and taking into account the growing algorithm of a specific type of mushroom.

A 24 h system test duration was chosen to make sure that all parts of the system worked properly (that the system was fully functional without errors and ready for the user), but the entire period of the mushroom growth cycle was not monitored, since the algorithm of the growth cycle of a specific mushroom species is entered in the control application by the final system user with the necessary education in the field of mushroom cultivation (e.g., an agronomist supervising the growth process, etc.). When testing the system's work, the monitoring of environmental parameters (temperature, humidity and $CO_2$) over time was chosen, as this is the most relevant for the user of the system in order to ensure automatic control of the mushroom cultivation cycle.

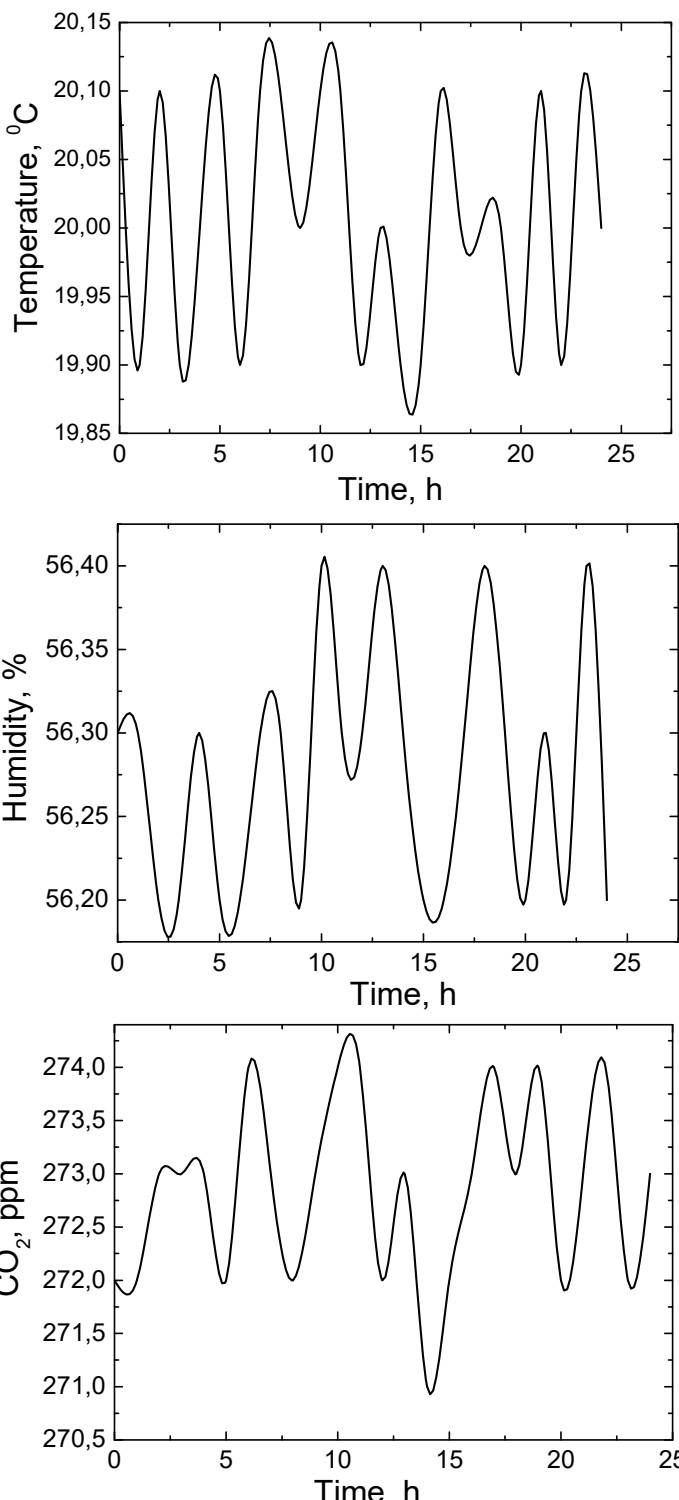

**Figure 12.** Changes in temperature, humidity and $CO_2$ in a mushroom growing room over a 24 h period.

Temperature, humidity and $CO_2$ levels were monitored and controlled during mushroom cultivation. Figure 13 shows the results of temperature, humidity and $CO_2$ control during the whole growth cycle of Shiitake mushrooms. When growing Shiitake mushrooms, it is important to maintain the ambient temperature in the range of 19–22 °C; the humidity must be maintained in the range of 80–83%, while during growth it is important to remove the amount of $CO_2$ additionally formed gas that is released by the growing mushrooms.

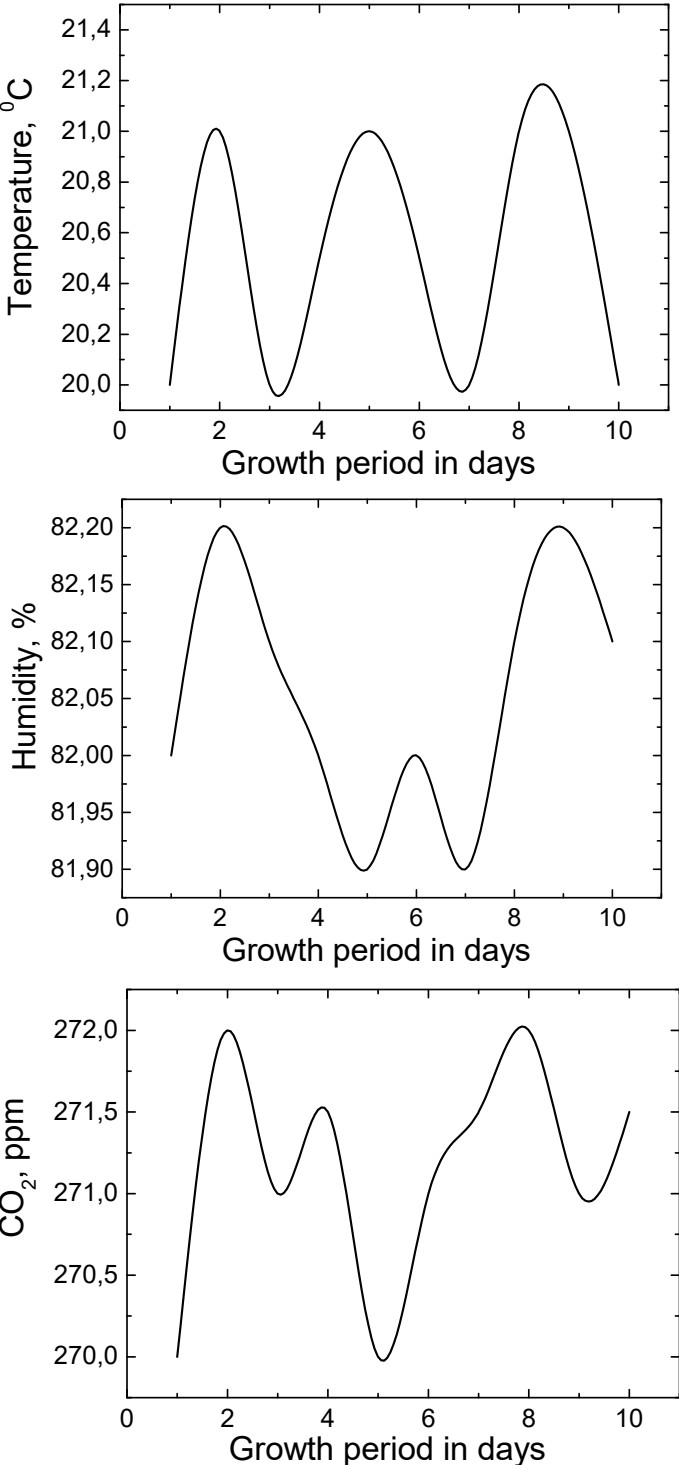

**Figure 13.** Changes in temperature, humidity and $CO_2$ parameters during the growth cycle of Shiitake mushrooms.

It is important that the $CO_2$ level is maintained as it was before the mushroom cultivation process. As we can see from Figure 13, the electronic mushroom cultivation system well ensured the maintenance of environmental parameters throughout the growth cycle. There were no deviations of the parameters from the technological requirements for the cultivation of this type of fungi. The complete process of growing mushrooms was carried out in automatic mode (control of environmental parameters was supported by the developed electronic system).

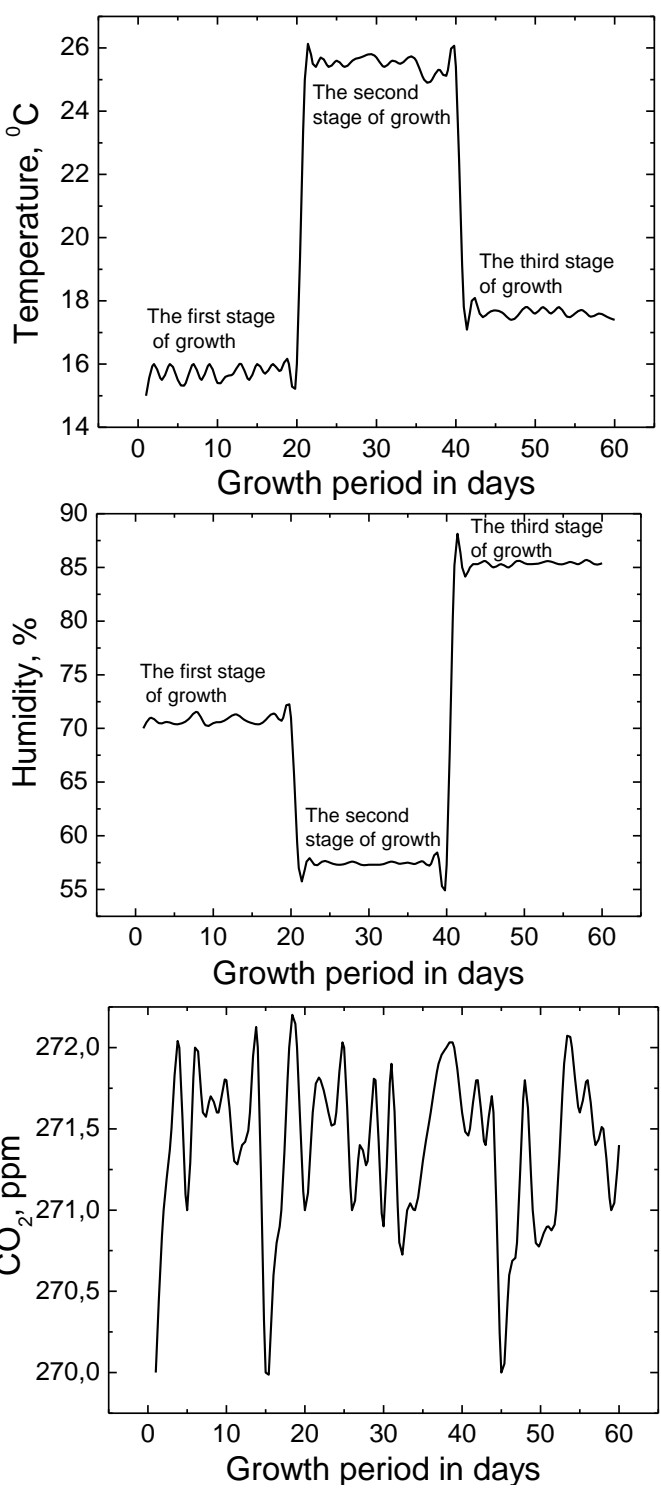

**Figure 14.** Changes in temperature, humidity and $CO_2$ parameters during the growth cycle of champignons.

Champignons were grown using the developed environmental parameters control system. The process of growing champignons consists of three stages and lasts about 60 days. In the first stage, it is necessary to maintain an ambient temperature of 14–16 °C, 71–73% humidity. For the second stage, a temperature of 24–26 °C and a humidity of 55–60% should be maintained. Meanwhile, for the final third stage of cultivation, the temperature must be kept within 16–18 °C and the humidity 85–90%.

The $CO_2$ level must be left as it was before the cultivation process, and any additional $CO_2$ must be removed. Figure 14 shows the results of maintaining environmental parameters for temperature, humidity and $CO_2$ level. All three growing stages were controlled automatically using the developed environmental parameters control system. As can be seen, by analyzing Figure 14 presented results during the entire growth cycle, the values of the environmental parameters did not exceed the permissible limits defined by the requirements for the cultivation of champignons.

Each type of mushroom requires different environmental conditions. It is important to maintain these conditions during the growth stages of the respective species of mushrooms. Often, when growing the respective species, the cycle of the mushroom growth process consists of several stages that require different environmental parameters (temperature, humidity, etc.). Consequently, at different stages of mushroom growth, operator intervention is required to adjust environmental parameters accordingly, which is inconvenient. In the application of our developed system, it is possible to enter the algorithm of the entire growth cycle of mushrooms with environmental parameters, which will be supported automatically even if the growth cycle consists of separate stages requiring different environmental parameters. In this case, additional operator intervention becomes unnecessary.

## 4. Conclusions

On the basis of microcontrollers, a monitoring and control system of environmental parameters of mushroom cultivation was developed. The main logic controller for environmental parameter monitoring consists of two levels: one level is for communication and receiving information from application sensors (communication board) and the other level is for information analysis and control of environmental conditions. Environmental parameters were monitored using $CO_2$, temperature and humidity modules (sensors). Appropriate devices such as air conditioners, fans, etc., are used to ensure the values of the environmental parameters based on the information received from the sensors. The control of these devices is performed by the main microcontroller, which is also used for the evaluation of sensor information. For each sensor, a printed circuit board was designed and manufactured separately, which are designed to transmit information to the MCU communication board. Since there are objects located in remote places and it is difficult to ensure a sufficient level of Wi-Fi connection between the individual components of the system, a local data transmission network based on the RS 485 interface and a pair of twisted wires were used in the production of this system prototype. To control the system, an optimal program algorithm was created, which allows to quickly and efficiently both receive and analyse information from environmental sensors, as well as to react quickly enough to deviations of environmental parameters. Attempting to make the work of the system as efficient and convenient as possible, a system management application was created using the Delphi programming language that allows the operator to quickly assess the progress of the process. In order to make the work as efficient as possible and to avoid human factor errors, the environment parameters support system also program-matically provides the "auto" work mode. These systems for maintaining and controlling the environmental parameters can be universal and can be adapted to the cultivation of any type of plant after the environmental control parameters and their limits, according to the conditions of plant cultivation, have been entered. In addition, if necessary, the capabilities of this prototype system can be expanded by adding environmental monitoring sensors and environmental control devices as required by the demand. The system was run in test mode and parameters were monitored for 24 h. As the results show, the control of application parameters using temperature, humidity and $CO_2$ sensors is stable and the deviations from the norm are permissible and do not exceed the deviations of the permissible values specified in most mushroom cultivation algorithms. During the process of growing mushrooms, both in the case of Shiitake mushrooms and in the case of champignons, environmental conditions were controlled in automatic mode. When growing both one and the other kind

of mushrooms, the values of the environmental parameters were maintained during the entire cultivation cycle and there were no deviations from the cultivation requirements.

The electric power of the entire mushroom cultivation system, which consists of a sensor system, a main controller, a data transmission network and devices supporting environmental parameters (fans, air conditioners, etc.) is about 3 kW. It is necessary to ensure a stable power supply for this system, since the mushroom growing process directly depends on the stability of the system. Part of the energy can be supplied by using wind or solar power as an additional source, using hybrid power systems. It is not recommended to power the mushroom cultivation system using only renewable energy sources, as the probability of energy supply instability increases, which directly affects the mushroom growth process.

**Author Contributions:** Conceptualization, Ž.K. and I.Š.; methodology, G.G.; software, I.Š.; validation, V.Č., G.G. and Ž.K.; formal analysis, I.Š.; investigation, Ž.K.; resources, G.G.; data curation, V.Č.; writing—original draft preparation, Ž.K.; writing—review and editing, G.G.; visualization, I.Š.; supervision, V.Č.; project administration, G.G.; funding acquisition, V.Č. All authors have read and agreed to the published version of the manuscript.

**Funding:** This research received no external funding.

**Informed Consent Statement:** Not applicable.

**Data Availability Statement:** Not applicable.

**Conflicts of Interest:** The authors declare no conflict of interest.

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
