# Peer review of "Intelligent Control of Mushroom Growing Conditions Using an Electronic System for Monitoring and Maintaining Environmental Parameters"

_applsci, doi:10.3390/app122413040_

Round 1
Reviewer 1 Report (Previous Reviewer 3)
I reviewed this manuscript when the authors submitted it to 'Sensors'. In my report I made it clear that the electronics were not the novel part of the work, as they are just routine use of standard components, but the application would be of interest if properly described. They need to explain what are the important parameter values at the different stages of mushroom growth and show how their system is achieving the correct control. How these parameters are different for different types of mushroom would be good. Photographs of the mushrooms growing at the different stages under the system control and the output from the control system over a full growing cycle. Without this, the paper lacks scientific originality and insufficiently rigourous.
Author Response
Please see the attachment

Reviewer 2 Report (New Reviewer)
The authors are constructing a system including devices and software capable of automating mushroom production environmental control.
I think that the description of the device and the information processing algorithm is sufficient, but I think that the novelty and attractiveness when compared with existing products should be further clarified.
In terms of the results, I think the manuscript needs to have a story that the results were obtained because of this control system, by comparing it with the existing system or without a control system.
Author Response
Please see the attachment

Reviewer 3 Report (New Reviewer)
- The manuscript describes a complete microcontrolled system to monitor and control temperature, humidity and CO2 levels in mushroom cultivation. The emphasis of the work is more technological than scientific. Authors make a detailed description of the equipment and software. In that sense, the manuscript reads more like a technical report than a journal article.
- In the "Introduction" section the authors mention the novelty of the work. I recommend that authors try to highlight the novelty. What is the gap in the literature that the authors intend to make a contribution to? What was the autors' insight?
- The keywords are very generic. I suggest replacing them with "smart cultivation control"; "electronic cultivation control"; "environmental parameters control"; "plant cultivation control"; "precision agriculture"; "environmental sensor monitoring"; "mushroom cultivation control".
- There are many syllable-diveded words throughout the text. Please review carefully.
Round 2
Reviewer 1 Report (Previous Reviewer 3)
I would like to thank the authors for taking on board my comments as it has taken the article from a description of some control electronics to being an interesting applied science article. I would suggest they double check the ppm of the CO2 as the quoted values are way below ambient air which is typically 450ppm. If their values are correct then this should be highlighted in the text and explained how it is achieved.
Author Response
Please see the attachment

Reviewer 2 Report (New Reviewer)
The authors have developed an automatic environmental control system to grow plants as economically and efficiently as possible.
As a result, CO2, temperature, humidity, etc. were controlled to the designated parameter levels, and the mushrooms were successfully grown efficiently.
The manuscript was adequately improved by revision.
This manuscript deserves publication.
Author Response
The reviewer did not provide any comments.
Reviewer 3 Report (New Reviewer)
The manuscript improved with the introduced modifications: the novelty is highlighted at the end of the "Introduction" and the good performance of the control system was demonstrated (results in Figs. 13 and 14).
Please rewrite the part "As we can see in Fig. The electronic mushroom growing system in Figure 13 is well secured...". There seems to be a writing error.
Round 3
Reviewer 1 Report (Previous Reviewer 3)
Active control of CO2 should be highlighted as a major part as in the text it just refers to measurement of CO2 rather than control. I think the authors response requires a bit more detail to be provided on the makes and models of the recuperation systems used as it would be essential for someone to repeat the process.
Author Response
Please see the attachment

Reviewer 3 Report (New Reviewer)
The authors implemented the reviewers' suggestions and I consider that the manuscript is now suitable for publication.
Author Response
The reviewer did not provide any comments.
This manuscript is a resubmission of an earlier submission. The following is a list of the peer review reports and author responses from that submission.
Round 1
Reviewer 1 Report
I tank the Authors for the poit-to-point answers to my questions.
Reviewer 2 Report
Compared to the first submission, the authors improved their manuscript including some measurement results and data about the power consumption. However, I don't think that the paper is mature enough to be considered for publication.
In the first revision round, I raised a major concern about the lack of novelty in this work. To address this issue, the authors add the following paragraph in the revised manuscript: “The novelty of this work is that individual parts of such a system (sensors, devices supporting environmental conditions, etc.) can transmit data to each other using different types of data transmission networks (RF modules, local area network, network of RS485 modules, Internet of Things, etc.) depending on the specifics of the object, location and conditions that allow the best to ensure stable data transmission and parameter control”.
I'm not sure how this can represent the novely of this work. It is well knoen that individual parts of the systems can communicate in different ways. Actually, it would have been interested a survey between the different approaches. In this work, instead, only a network of RS485 has been adopted, and this does not represent a novel contribution to the state-of-the-art.
Concerning the measurements, it's not evident from them how the proposed system could be better than other solutions. Apart from showing the functionality of the system in a single scenario and for a single time record, there is nothing more. For example, a comparison of the same physical quantities measured with and without the control system would help proving the goodness of the proposed system.
About the comparison with the state-of-the-art, finally, nothing has been really added.
Reviewer 3 Report
This paper is well written and clearly describes the development of an electronic control system. The description of the system is very clear and has sufficient, if not, too much detail included. I do not feel that the 24 hours of collected data represents sufficient to demonstrate claimed novelty of being bespoke for mushroom growing. I am left questioning how long the growing cycle takes and what are the various parameter requirements through this period. It would then be good to demonstrate how the control system has responded to this specific need. The system is not 'cutting edge' being hard wired and running from mains power so this can't be claimed part of the novelty. I encourage that a focus on the specific requirements of mushrooms is made and a comprehensive set of data showing switching on and off times of the control systems in response to the sensors to deliver this. Some photographs of the mushrooms growing would allow a sense of scale to be understood.
Round 2
Reviewer 2 Report
Compared to the previous version, the authors basically improved the introduction of their manuscript, highlighting the novelty of the proposed work (after my suggestion). However, I don't understand how it is possible that the novelty of this work changed from the previous revision round to the actual one. In the previous version, in fact, they stated: "The novelty of this work is that individual parts of such a system (sensors, devices supporting environmental conditions, etc.) can transmit data to each other using different types of data transmission networks (network of RF modules, local network, network of RS485 modules, Internet of Things, etc.) depending on the specifics of the object, location and conditions that allow the best to ensure stable data transmission and parameter control."
Now, the new claim is:
"The novelty of this work is that was created unique program algorithm and system control software code with a control application (located on the control panel in computer or central server) that allows the user to enter the parameters of the growing conditions of various types of mushrooms. Such a programming solution allows for automatic maintenance of environmental conditions during the entire mushroom growth cycle. Automatic control takes place according to the information received from environmental parameters sensors."
This change, however, is not reflected in the whole body of the revised manuscript, because nothing is really changed. If the main novely of this work is represented by the control algorithm, more details about the code should be provided, as well as an extensive comparison with the state-of-the-art in terms of control algorithm.
The other doubts that I raised in the previous revision round were not really answered.
Reviewer 3 Report
Sadly, the authors have not understood that this manuscript lacks sufficient novelty for publication as a 'universal control system' as such systems have been around for decades. The specific environmental requirements for mushroom growth and how this system specifically addresses this, with examples, is what would add the novelty. They have failed to add the specific details that show the particular demands of mushroom growth and how their system addresses this. I would encourage them to reread my previous comments and try to address them.